Cytological and molecular characterizations of a novel 2A nullisomic line derived from a widely-grown wheat cultivar Zhoumai 18 conferring male sterility

Jiao Zhixin 1
Zhu Xinxin 1
Li Huijuan 1
Liu Zhitao 2 3
Huang Xinyi 2
Wu Nan 2
An Junhang 1
Li Junchang 1
Zhang Jing 1
Jiang Yumei 1
Li Qiaoyun 1
Qi Zengjun 2
Niu Jishan jsniu@henau.edu.cn 1
1 Henan Agricultural University, National Centre of Engineering and Technological Research for Wheat / National Key Laboratory of Wheat and Maize Crop Science , Zhengzhou , Henan , China
2 Nanjing Agricultural University, State key Laboratory of Crop Genetics and Germplasm Enhancement , Nanjing , Jiangsu , China
3 Sichuan Academy of Agricultural Sciences, Crop Research Institue , Chengdu , Sichuan , China
Sun Genlou
Electronic publication date: 2020 Oct 30
Publication date: 2020
Volume: 8
Electronic Location ID: e10275
Received 2020 Jun 19; Accepted 2020 Oct 8
Copyright: ©2020 Jiao et al.
Copyright year: 2020
Copyright holder: Jiao et al.
License: This is an open access article distributed under the terms of the Creative Commons Attribution License, which permits unrestricted use, distribution, reproduction and adaptation in any medium and for any purpose provided that it is properly attributed. For attribution, the original author(s), title, publication source (PeerJ) and either DOI or URL of the article must be cited.
License URL: https://creativecommons.org/licenses/by/4.0/

Keywords: Wheat (Triticum aestivum L.), dms, Chromosome, Nullisomic line, Pollen development

Funding: National Natural Science foundation of China 31571646 key project of Co-construction State Key Laboratory of Wheat and Maize Crop Science in 2020 This study was supported by the National Natural Science Foundation of China (NSFC, 31571646) and the Key Project of Co-construction State Key Laboratory of Wheat and Maize Crop Science in 2020. The funders had no role in study design, data collection and analysis, decision to publish, or preparation of the manuscript.

==============================
A dwarf, multi-pistil and male sterile dms mutant was previously reported by us. However, the genetic changes in this dms are unclear. To examine the genetic changes, single nucleotide polymorphism (SNP) association, chromosome counting, and high-resolution chromosome fluorescence in situ hybridization (FISH) techniques were employed. By comparing tall plants (T) with dwarf plants (D) in the offspring of dms mutant plants, SNP association analysis indicated that most SNPs were on chromosome 2A. There were three types in offspring of dms plants, with 42, 41 and 40 chromosomes respectively. High-resolution chromosome painting analysis demonstrated that T plants had all 42 wheat chromosomes; the medium plants (M) had 41 chromosomes, lacking one chromosome 2A; while D plants had 40 wheat chromosomes, and lacked both 2A chromosomes. These data demonstrated that dms resulted from a loss of chromosome 2A. We identified 23 genes on chromosome 2A which might be involved in the development of stamens or pollen grains. These results lay a solid foundation for further analysis of the molecular mechanisms of wheat male sterility. Because D plants can be used as a female parent to cross with other wheat genotypes, dms is a unique germplasm for any functional study of chromosome 2A and wheat breeding specifically targeting genes on 2A.

Introduction

Allohexaploid wheat (Triticum aestivum L., 2n = 6x = 42, genomic constitution AABBDD) has a genome from three diploid species: Triticum urartu Thum. (the source of the A genome), Aegilops speltoides (Tausch) Gren. or a closely related species (the source of the B genome), and Aegilops tauschii Coss. (the source of the D genome) (Huang et al., 2002). Because allohexaploid wheat has high level of functional redundancy, it can host a range of diverse whole-chromosome aneuploids (Zhang et al., 2013; Zhang et al., 2017). There are various types aneuploid variations available in wheat, such as nulli-tetrasomic lines, nullisomic lines, monosomic lines, ditelosomic lines, as well as chromosome fragmental deletion lines (Endo & Gill, 1996; Qi et al., 2003). The aneuploid stocks are useful in wheat gene mapping, and especially, genes can be located to a small segment using a series of chromosome fragmental deletion lines (Endo & Gill, 1996; Qi et al., 2003). A set of homozygous chromosome deletion lines were obtained in cv. Chinese Spring (Endo & Gill, 1996). Most of the homozygous chromosome 2A short arm deletion lines are sterile; genes involved in male development are located on the chromosome 2A short arm (Endo & Gill, 1996).

Presently, at least five stable genic male sterility (GMS) genes are known in bread wheat. They are ms1 on 4BS (Wang et al., 2017), Ms2 on 4DS (Xia et al., 2017), Ms3 on 5AS (Maan et al., 1987), Ms4 on 4DS (Maan & Kianian, 2001) and ms5 on 3AL (Pallotta et al., 2019), but the molecular regulation mechanisms of the male sterility lines are largely unknown.

Anther and pollen developments are complex biological processes, determining wheat male fertility. Pollen development starts from a single cell layer, which undergoes a series of cell divisions and differentiation to form the innermost meiocytes encased within four somatic anther cell layers; from inner to outer these are the tapetum, middle layer, endothecium and epidermis (Zhang & Yang, 2014). The tapetum serves as a nutritive tissue, providing metabolites, nutrients, and cell wall precursors for the development of pollen grains (Goldberg, Beals & Sanders, 1993). The regulatory genes involved in pollen exine patterning are known in Arabidopsis and rice (Pearce et al., 2015; Lin et al., 2017). For example, AtMS1 (Male Sterility 1) (Wilson et al., 2001), AtDRL1 (Dihydroflavonol 4-Reductase-Like 1) (Tang et al., 2009), AtLAP3 (Less Adherent Pollen 3) (Dobritsa et al., 2009), AtLAP5 (Less Adherent Pollen 5) (Dobritsa et al., 2010) in Arabidopsis, and OsGAMYB (GA, gibberellin; MYB, v-myb avian myeloblastosis viral oncogene homolog) (Aya et al., 2009), OsNP1 (No Pollen 1) (Liu et al., 2017), OsDPW2 (Defective Pollen Wall 2) (Xu et al., 2017) in rice are involved in the biosynthesis of sporopollenin, which is a major constituent of exine in the outer pollen wall. However, pollen developmental studies in wheat have lagged behind these plants.

Previously, we reported a mutant of dwarf, multi-pistil and male sterile dms in wheat (Duan et al., 2015; Zhu et al., 2016). Pollen vigor and hybridization tests demonstrated that dms mutant was male sterile. Male sterility and male fertility followed a segregation ratio of 1:3 [D:(T +M) = 1:3] (Duan et al., 2015). However, the genetic basis of this mutant is unknown. The present study is to discover the genetic basis at the cytological and molecular levels. We also identified a set of genes playing potential key roles accounting for male sterility on chromosome 2A.

Materials & Methods

Plant materials

Mutant dms used in this study was obtained as previously described by us (Duan et al., 2015). All plant materials were planted in our experimental field at Houwang Village, Xingyang City, Henan, P. R. China (34°51′N, 113°35′E, 49 m above sea level). We have got permission prior to accessing the field site by Henan Seed Company, and the administrator’s name is Haolong Yang.

Morphological and histochemical observations

Phenotype of plants, young spikes and stamens of dms were observed at different developmental stages (Zadoks, Chang & Konzak, 1974). Developmental stages of the young spike primordia were referenced as previously described (Vahamidis et al., 2014). The florets with anthers and pistils at different developmental stages were dissected from spikes using an anatomical needle. Anthers were observed using scanning electron microscope (SEM) (SU8010, Hitachi, Tokyo, Japan) as previously described (Jiao et al., 2019). Structural and histochemical observation on the anthers of T and D plants were carried out using a method previously described by Geng et al. (2018).

RNA extraction and mRNA sequencing

Young spikes of T, M and D plants at floret primordium visible stage (stage 7) (Vahamidis et al., 2014) and anthers of T and D plants at 3-nucleate pollen stage (stage 12) (Browne et al., 2018) were sampled for RNA extraction. All the five samples for mRNA sequencing had 3 biological replicates. Total RNAs of fifteen samples were extracted respectively with TRIzol® reagent (TransGen Biotech, Beijing, China). RNA sequencing and basic analysis were carried out in BioMarker Company (Beijing, China).

mRNA data analysis

The mRNA reads of the fifteen samples were aligned to the draft assembly (IWGSC v1.0) of the wheat genome survey sequence (http://www.wheatgenome.org/) using Tophat2 tool software (Kim et al., 2013). Reads distribution on 21 wheat chromosomes of T, M and D plants were analyzed and drawn using Circos tool software (http://mkweb.bcgsc.ca/tableviewer/). Gene functional annotation was carried out as previously described by us (He et al., 2018). Gene expression levels were estimated by fragments per kilobase of transcript per million fragments mapped (FPKM) (Florea, Song & Salzberg, 2013). Differentially expressed genes (DEGs) between two sample groups were analyzed using DESeq R package (Wang et al., 2010). The FDR < 0.01 (false discovery rate) and FC ≥ 2 were set as the thresholds for significantly DEGs.

Principal components were calculated using the FPKM values (Schulze et al., 2012). For the single gene, heat maps were drawn using software Hem I v 1.0.3.7 according to the FPKM values. Tissue specific expression patterns of wheat genes were analyzed using k-means cluster on the BMKCloud platform (https://www.biocloud.net/). Clustering was performed in R using the k-means function, where k = 8 within the cluster package by Euclidean distance. Wheat tissue type, expression profile data sources and sample description in origin article (Garcia-Seco et al., 2017; Feng et al., 2017; Li et al., 2019) were listed in Table S1.

Single nucleotide polymorphism (SNP) association analysis

Clean reads data of the two bulks (young spikes of T and D plants) were selected for SNPs identification. After the reads were aligned to the Chinese Spring genome, the SNPs were called using module ‘HaplotypeCaller’ software GATK v3.6 (McKenna et al., 2010). Euclidean distance (ED) algorithm was used to calculate the grade of different SNPs between T and D plants (Hill et al., 2013).

Simple sequence repeat (SSR) analysis

Genomic DNAs of T, M and D plants were extracted from young leaves using cetyltrimethyl ammonium bromid (CTAB)-method (Clarke, 2009). DNA concentration was measured with a DU800 Nucleic Acid Protein Analyzer (Beckman coulter, Fullerton, California, USA). SSR markers distributed on wheat chromosome 2A were used to analyze the genetic polymorphisms among T, M and D plants. SSR primers employed in this study included those of GWM (Röder et al., 1998), WMC (Somers, Isaac & Edwards, 2004), BARC (Song et al., 2005) and GPW (Gupta et al., 2002). PCR amplification reactions were performed using a method previously described by Zhu et al. (2019). PCR products were separated on an 8% PAGE gel with a standard DNA molecular weight marker in the first lane of the gel. The products were run at 60 W for about 1.0 h. Then the gel was removed from the apparatus and stained using the silver staining method (Li et al., 2018b). Primers used for SSR analysis are list in Table S2.

Cytological analysis

Chromosome configurations and high-resolution chromosome painting analysis of plants derived from M plants at metaphase of mitosis. Chromosome samples were prepared as previously described (Du et al., 2017). For chromosome painting, eight single strand oligonucleotides were used to form a modified multiplex probes for karyotype analysis in wheat, which included TAMRA (6-carboxytetramethylrhodamine)-modified oligonucleotides pAs1-1, pAs1-3, pAs1-4, pAs1-6, AFA-3 and AFA-4, and two FAM (6-carboxyfluorescein)-modified oligonucleotides pSc119.2-1 and (GAA)10. Oligonucleotide probes used for FISH are list in Table S3. The FISH procedure was tested as previously described (Du et al., 2017). Chromosomes were visualized with microscope Olympus BX51 and pictures were captured with SPOT CCD (SPOT Cooled Color Digital Camera). Image analysis was conducted using Photoshop v6.0.

qRT-PCR for mRNAs

Anthers of T and D plants at 3-nucleate pollen stage were prepared for real-time PCR. The experimental samples were consistent with the samples of RNA-seq. The qRT-PCR was performed as previously described by us (He et al., 2018). Reverse transcription was performed with 1 µg RNA using Hifair® II 1st Strand cDNA Synthesis SuperMix (11123ES60, Yeasen, Shanghai, China). Real-time PCRs of mRNAs were performed using Hieff® qPCR SYBR Green Master Mix (1201ES08, Yeasen, Shanghai, China) and a CFX ConnectTM Real-Time System (Bio-Rad, Hercules, CA, USA) following the production instructions. The wheat actin gene was used as an internal control. All primer sequences are listed in Table S4. The gene expression levels were calculated according to the 2−ΔΔCt method (Livak & Schmittgen, 2001). The SPSS version 17.0 software (SPSS Inc., Chicago, IL, USA) was applied for statistical analysis for the qRT-PCR.

Results

Male abortion in D plants of dms

There were three typical phenotypes in the progeny of dms, tall (T), semi-dwarf (M) and dwarf (D) plants (Fig. S1). D plants were male sterile according to the result of I2–KI staining (Duan et al., 2015). To discover the morphological causes of the male abortion in D, scanning electron microscope was employed to observe the anther ultrastructure (Fig. 1). Unlike the fertile T plants, the anthers of D plants were not dehiscent at the trinucleate stage, and no mature pollen grains were released (Figs. 1A and 1B). The outer anther epidermal cells of T plants were arranged neatly, whereas these of D plants were irregular (Figs. 1C and 1D). Moreover, compared with the anthers of the T plants, the inner epidermal of D plants showed aberrant sized Ubisch bodies, which suggested a structural abnormality (Figs. 1E–1H). The pollen grains of T plants were plump and rounded; by contrast, the pollen grains of D plants were extremely shriveled and atrophied (Figs. 1I and 1L). Furthermore, we performed a histological analysis of the anthers in the dms mutant and the wild type. In the initial stages of pollen development (Figs. S2A and S2B), there were no differences between the wild type and dms mutant. The young microspores were normal (Fig. S2C and S2D). However, compared with the wild type, degenerated pollen grains were observed in the dms mutant at 3-nucleate pollen stage (Figs. S2E and S2F).

Figure 1 The scanning electron micrographs of the anthers, anther epidermis, anther inner surface, and pollen grains in the T and D plants of dms at the trinucleate stage.

Uby: Ubisch bodies. Scale bars represent 1 mm in anthers, 500 µm in pollen, 50 µm on the epidermis surface and 10 µm on the inner surface.

The female organs in flowers of D plants developed unwell, most of the pistils were normal, a few were multi-pistils (Fig. S3). D plants were male sterile but female fertile, it can be used as a female parent to cross with other wheat genotypes. We got F1 seeds from dwarf plants (as female parent) crossed with Guomai 301 (as pollen parent) (Fig. S4).

The most SNPs were on chromosome 2A

In order to clarify the genetic changes in dms, SNP association analysis was carried out. A total of 176.73 Gb data were obtained from the five super bulked samples of dms by mRNA sequencing: young spikes of T plants (T-YS; T1a, T1b, T1c) (Fig. S5A), young spikes of M plants (M-YS; T2a, T2b, T2c) (Fig. S5B), young spikes of D plants (D-YS; T3a, T3b, T3c) (Fig. S5C), stamens of T plants (T-ST; T4a, T4b, T4c) (Fig. S5D), stamens of D plants (D-ST; T5a, T5b, T5c) (Fig. S5E). SNP association analysis indicated that the most SNPs between T and D plants were on chromosome 2A (Fig. 2). Total 523 SNPs were significantly different between T-ST and D-ST, ED>0.33, in them, 230 were on chromosome 2A, which occupied 44% of the total SNPs (Fig. S6; Table S5). The result implied that the mutation occurred on chromosome 2A. The results were consistent with the sequencing reads distribution on chromosome 2A of young spikes in T, M and D plants (Fig. 3). About 5.24% of the total reads was identified on chromosome 2A in T plants, which was about twice that of M plants (2.86%) (Table S6).

Figure 2 Chromosome distribution of the SNPs between stamens of T and D plants.

ED is Euclidean distance. Each dot represents a SNP identified between T and D. The serial numbers of wheat chromosomes are indicated on the X-axis. The red dotted line indicates significance threshold (ED = 0.33). The wavy line above the significance threshold indicates the variant section.

Figure 3 Circular diagram of the reads distributed on the 21 wheat chromosomes of T, M and D plants.

The tracks toward the center of the circle display (A) chromosome name and size (Mb); (B) read segments density of D plant; (C) read segments density of M plant; (D) read segments density of T plant. The triangles highlight reads distributions on chromosome 2A.

SSR markers on chromosome 2A were polymorphic among tall, semi-dwarf and dwarf plants

In order to further verify that the genetic variation of dms occurred on chromosome 2A, eleven plants including three typical phenotypes of tall, semi-dwarf and dwarf were randomly selected for SSR genotyping. SSR markers evenly distributed across wheat A, B and D genomes were used to detect the polymorphism between tall and dwarf plants of dms. The result showed that all the primers specifically amplifying fragments on chromosomes 2A (Xgwm312, Xgpw2229, Xgwm95, Xgwm445, Xwmc794, Xgwm328, Xgwm425, Xbarc212 and Xbarc122) couldn’t amplify the expected products in the dwarf plants of dms (Fig. 4; lane 4, 7, 9, 11), which implied that wheat chromosome 2A was missing in the dwarf plants.

Figure 4 SSR markers on chromosome 2A were polymorphic.

M, DNA molecule weight markers; lanes 1–11, the offspring of dms mutant. lanes 1, 3, 5 and 8 were tall plants; lanes 2, 6 and 10 were semi-dwarf plants; lanes 4, 7, 9 and 11 were dwarf plants. The arrows indicate the SSR products.

The SSR markers were used to quantitatively amplify the DNA templates of tall, semi-dwarf and dwarf plants. The amounts of the amplified products of the tall plants (Fig. 4; lane 1, 3, 5, 8) were about double that of the semi-dwarf plants (Fig. 4; lane 2, 6, 10), and dwarf plants without PCR products (Fig. 4). In another word, the SSR primers specifically amplifying fragments on chromosomes 2A could be used to distinguish the T, M and D plants in the progeny of dms.

Chromosome 2A was absent in D plants

To explore the chromosome number of dms, 42 plants derived from M plants were investigated. Among them, 17 plants had 42 chromosomes, 21 had 41 chromosomes and 4 had 40 chromosomes (Table 1). After chromosome analysis, the seedlings were planted in field. The 17 plants with 42 chromosomes showed regular plant height, pistil and fertility, the 21 plants with 41 chromosomes showed medium plant height and regular pistil and fertility, while all the 4 plants with 40 chromosomes showed dwarf status, multi-pistil and complete male sterility (Fig. S7).

Table 1 Chromosome numbers and phenotypes of the progeny derived from M plants.

Seedling number detected	Chromosome number	Phenotype	
17	42	Normal tall plant (T)	
21	41	Semi-dwarf plant (M)	
4	40	Dwarf, multi-pistil and male sterility (D)	

To validate and identify the chromosome constitution of dms, high-resolution chromosome painting was applied using eight single strand oligonucelotide probes. The result showed that all 21 wheat homoeologous chromosome pairs can be reliably discriminated. Total 45 plants derived from M plants were clearly characterized, among them, 18 plants had 42 chromosomes and regular karyotypes (Figs. 5A and 5B), 20 plants had 41 chromosomes and lacked one chromosome 2A (Figs. 5C and 5D), while the remaining 7 plants had 40 chromosomes and lacked a pair of chromosomes 2A (Figs. 5E and 5F).

Figure 5 High-resolution chromosome painting analysis of the plants derived from M plants at metaphase of mitosis.

(A) and (B), one karyotype of T plant, which had 42 normal chromosomes. (C) and (D), one karyotype of M plant, which had 41 chromosomes, lacked one chromosome 2A. (E) and (F), one karyotype of D plant, which had 40 chromosomes, lacked a pair of chromosomes 2A. Blue color, chromosomes counterstained with DAPI; green, signals of oligos pSc119.2-1, (GAA)10; red, signals of oligos AFA-3, AFA-4, pAs1-1, pAs1-3, pAs1-4 and pAs1-6. Scale bar = 10 µm.

Chromosome 2A carries key genes determining fertility

We assessed the effects of several different chromosome 2As by making six crosses (Table 2). At F1 generation, all six crosses segregated into two phenotypes, tall plants (T) and Semi-dwarf plants (M). Combined with the data above, we knew that the genotypes of the cross Zhoumai 18 × M at F1 generation were 2Azhoumai182Adms and 2Azhoumai18, the corresponding phenotypes were T and M plants. This demonstrated that the transmitting frequency of null-2A pollens was about half of the 2A pollens (Table 2). All the T plants of the six crosses at F1 generation didn’t segregate at F2 generation, but all M plants at F1 generation segregated into T, M and D plants at F2 generation. All M plants at F2 also segregated into T, M and D plants at F3 (Table 2). The 2As played key roles during whole plant development in all the monosomic lines of 2As from Taishan 4429, Jing 08-426, Yangmai 11, Yuyou 1 and Guomai 301. These data demonstrated that the six different origin chromosomes 2As had the similar function as the chromosome 2A in dms. In addition, the heterozygous genetic backgrounds of the six crosses, including chromosomes 2Ds and 2Bs couldn’t complement the effects of chromosome 2A. D plants were used as a female parent to cross with other wheat genotypes, so as to construct inter-cultivar chromosome 2A substitution lines (Fig. S8). At F1 generation, all the lines had one chromosome 2A, they were male fertile, which indicated chromosome 2A carries key genes determining male fertility.

Table 2 Phenotypes of the six crosses at F1 and F2 generations.

Cross.	F1a	F2b	
Female	Male (M plants of dms)	Phenotype and observed lines (number)	Phenotype and observed lines (number)	
Zhoumai18	M	T (22)	T			
		M (10)	T(12)	M(23)	D(10)	
Taishan 4429	M	T (3)	T			
		M (6)	T(10)	M(21)	D(9)	
Jing 08-426	M	T (3)	T			
		M (3)	T(8)	M(15)	D(6)	
Yangmai 11	M	T (3)	T			
		M (6)	T(13)	M(26)	D(12)	
Yuyou 1	M	T (3)	T			
		M (6)	T(15)	M(30)	D(14)	
Guomai 301	M	T (3)	T			
		M (6)	T(12)	M(24)	D(9)	
Notes.

a At F1 generation, all six cross combinations segregated into two phenotypes, the total number of T plant was 37, the total number of M plant was 37.

b All seeds harvested from M plants at F2 generation will segregate into T, M and D plants at F3 generation as dms.

Stamen-specifically expressed genes on chromosome 2A

To identify the key genes determining fertility in wheat, 5939 genes located on chromosome 2A (TraesCS2A01G000100 - TraesCS2A01G593900) were analyzed. Total 4613 genes on chromosome 2A were identified and found to be homeologous to genes on other chromosomes (Table S7). Deletion of these genes on chromosome 2A might not affect the phenotype of D plants because of gene redundancy. Total 1326 2A-specific genes were identified (Table S8). Tissue specific expression analysis showed that 188 genes on chromosome 2A were stamen specifically expressed genes (Fig. 6A; Table S9). Venn diagram analysis showed that 23 genes were chromosome 2A specific and stamen specifically expressed genes (Fig. 6B; Table S9). Among them 6 genes were reported to be involved in pollen development related biological processes (Table 3), they may be the key genes determining male fertility.

Figure 6 Chromosome 2A carries key genes involved in pollen development.

(A) The expression profiles of the stamen specifically expressed genes on 2A. (B) Venn diagram showed the chromosome 2A and stamen specifically expressed genes.

Table 3 The key genes determining male fertility in wheat on chromosome 2A.

ID	Homologous gene/ Gene name	Protein product	Reported function / Mutant phenotype	Reference	
TraesCS2A01G043400	AT5G40260/AtSWEET	SWEET sugar transporter	A role in glucose efflux for pollen nutrition / Male sterility	Chen et al. (2010)	
TraesCS2A01G051800	AT5G40260/AtSWEET	SWEET sugar transporter	A role in glucose efflux for pollen nutrition / Male sterility	Chen et al. (2010)	
TraesCS2A01G261500	At4g05330/AtAGD13	C2H2 transcription factor	Involved in pollen development / Disrupted pollen cell wall	Reňák, Dupl’áková & Honys (2012)	
TraesCS2A01G442500	Os01g0293100/OsTIP2	bHLH transcription factor	Control of anther cell differentiation / Male sterility	Fu et al. (2014)	
TraesCS2A01G521900	At5g49180/AtPME58	A putative pectin methylesterase	Involved in pollen germination and tube growth	Leroux et al. (2015)	
TraesCS2A01G527700	At5g13930/AtCHS	Chalcone synthase	Involved in flavonoid biosynthesis	Dobritsa et al. (2010)	

DEGs involved in pollen development related signal transduction in dms

A total of 5199 genes were significantly differentially expressed (FC ≥ 4) between stamens of T and D plants. Among them 4761 DEGs expressed less, only 438 DEGs expressed highly in D-ST (Table S10). Obviously, the expressions of most DEGs in D plants of dms were lower because lacking of the chromosome 2A. We identified 229 putative TF DEGs between T-ST and D-ST. They belonged to 47 TF families. The top three with the most number of DEGs were MYB, C2H2 zinc finger protein (C2H2) and APETALA 2/ethylene-responsive element binding factor (AP2/ERF) transcription factor families and most of them expressed less in D-ST (Table S11).

Totally 45 DEGs were associated with auxin homeostasis, such as biosynthesis, response, signaling and metabolism (Fig. S9). Among auxin signal transduction-related genes, homologs of auxin response factor (ARF) genes and several auxin biosynthesis-related genes expressed less in D-ST. Reduced expression of auxin biosynthesis and signal transduction related genes were closely related to male sterility (Su et al., 2019). Abnormality of auxin homeostasis might be a major factor leading to the phenotype of dms.

DEGs involved in pollen development related metabolism in dms

Enrichment analysis of the DEGs between T-ST and D-ST revealed that most of them were involved in pollen development related metabolism such as the GO terms of ‘lipid metabolism’, ‘cell wall biogenesis’ and ‘pollen development’ (Fig. 7). Among lipid metabolism related genes, fatty acid biosynthetic related genes expressed less and lipid catabolic process related genes expressed highly in D-ST. Among cell wall biogenesis related genes, cell wall modification related genes expressed less and cellulose biosynthetic process related genes expressed highly in D-ST (Fig. 7). This indicated that fatty acid metabolic and cell wall assembly disorder might be the critical factors causing male sterility in dms.

Figure 7 The DEGs of GO category in lipid metabolism, cell wall biogenesis and pollen development between stamens of T plants and D plants.

Down, the genes were expressed less in stamens of D plants; Up, the opposite.

DEGs involved in abnormal pollen development of dms

The DEGs associated with various aspects of pollen development, such as pollen tube growth, pollen germination, megagametogenesis, anther dehiscence and anther wall tapetum formation, expressed less in D-ST (Fig. 7). A series of pollen development related genes were identified (Table 4). The homologs of AtSAC1B and OsGAMYB involved in exine formation and AtAGC1.5, OsSUT1, AtRopGEF8, AtAGC1.7, AtFIM5 and etc. involved in pollen germination and pollen tube growth expressed less in D-ST (Table 4). The qRT-PCR results showed that the expression patterns of the representative genes were well consistent with that of the sequencing results (Fig. 8).

Table 4 Key genes involved in abnormal pollen development of dms.

ID	T-ST_FPKM	D-ST_FPKM	Homologous gene/ Gene name	Reported function	Reference	
*TraesCS1A01G020700	32	9	At5g66020/SAC1B	Involved in pollen exine formation	Despres et al. (2003)	
*TraesCS1B01G024700	2	0	
*TraesCS1D01G020200	31	9	
TraesCS1A01G187500	7	49	Os10g0524500/NP1	Required for male fertility	Liu et al. (2017)	
TraesCS1B01G195300	6	48	
TraesCS1D01G189200	8	53	
*TraesCS2A01G288400	300	0	Os04g0398700/PS1	Pollen specific protein	–	
*TraesCS2B01G305200	297	57	
*TraesCS2D01G286700	323	62	
*TraesCS4A01G000100	151	27	TaPhl p 5	Causes grass pollen allergy	–	
*TraesCS4D01G000100	130	24	
TraesCS5A01G233600	10	19	At5g22260/MS1	Required for male sterility	Wilson et al. (2001)	
TraesCS5B01G232100	7	18	
TraesCS5D01G240500	10	31	
TraesCS7A01G269700	1341	837	Os08g0546300/C4	Required for pollen wall development	Li & Zhang (2010)	
TraesCS7B01G167900	395	105	
TraesCS7D01G270200	353	88	
TraesCS7A01G458700	41	5	Os01g0812000/GAMYB	Required for tapetum function and pollen wall formation in rice	Aya et al. (2009)	
TraesCS7B01G357900	48	6	
TraesCS7D01G446700	53	7	
*TraesCS1A01G133200	15	1	AT3G12690/AGC1.5	Involved in polarized growth of pollen tubes	Li et al. (2018a)	
*TraesCS1B01G152700	11	1	
*TraesCS1D01G124200	5	3	
*TraesCS1A01G134100	26	4	Os03g0170900/SUT1	Essential for normal pollen germination	Hirose et al. (2010)	
*TraesCS1D01G135900	35	11	
*TraesCS3A01G157100	13	1	AT1G79250/AGC1.7	Involved in polarized growth of pollen tubes	Li et al. (2018a)	
*TraesCS3B01G183500	8	3	
*TraesCS3D01G164700	13	1	
*TraesCS3A01G270400	70	2	AT3G24620/RopGEF8	Required for pollen tube growth	Gu et al. (2006)	
*TraesCS3B01G304200	40	0	
*TraesCS3D01G270200	98	4	
TraesCS4A01G001100	9	0	AT3G04690/ANXUR1	Involved in premature pollen tube rupture	Boisson-Dernier et al. (2009)	
TraesCS4B01G002300	13	0	
TraesCS4D01G001000	3	0	
TraesCS4A01G185600	12	2	AT3G07960/PIP5K6	Regulates clathrin-dependent endocytosis in pollen tubes	Zhao et al. (2010)	
TraesCS4B01G133100	13	2	
TraesCS4D01G128000	13	2	
*TraesCS6A01G200700	13	2	AT5G35700/FIM5	Required for pollen germination and pollen tube growth	Su et al. (2017)	
*TraesCS6B01G226800	14	1	
*TraesCS6D01G189200	19	2	
TraesCS7A01G281300	232	53	Os08g0560700/Phl p 7	Involved in pollen tube growth	Verdino et al. (2002)	
TraesCS7B01G179800	277	66	
TraesCS7D01G280000	471	109	
Notes.

* The asterisks indicate the genes verified by real-time qRT-PCR.

Figure 8 Spatiotemporal expression profiles of the key genes involved in abnormal pollen development of dms.

(A) SAC1B, (B) PS1, (C) Phl p 5, (D) AGCL5, (E) SUT1, (F) AGC1.7, (G) RopGEF8, (H) FIM5. The asterisks indicate the significant difference between different samples at **P = 0.01. All qRT-PCR reactions were replicated thrice.

A hypothesis of the molecular regulatory network in dms wheat lines

In summary, we put forward a hypothesis on the molecular regulatory network in dms (Fig. 9). Several evidences support this hypothesis: (1) Chromosome 2A is absent in D plants (Fig. 4); (2) key genes involved in pollen development are identified on chromosome 2A in dms (Table 3); and (3) fatty acid biosynthetic related genes expressed less and catabolic related genes expressed highly in dms (Fig. 7).

Figure 9 A hypothesis on the molecular regulatory network in dms.

The modules coloured in light red are highly expressed in dms and modules coloured in light blue are expressed less in dms.

Discussion

Aneuploids are large scale mutations greatly affect cellular physiology and have significant phenotypic consequences (Zhang et al., 2017). The typical phenotype of dms was significantly different from its parent Zhoumai 18 at three traits, plant height, pistil number and male fertility. Preliminary, this mutant was considered as a SNP mutation. Till the SNP association analysis showed that a large amount of SNPs between T and D plants, we thought their chromosomes should be clarified. High resolution chromosome painting is a new and efficient method for distinguishing chromosomes, which has many advantages including high sensitivity and resolution (Du et al., 2017). Using this method, we successfully distinguished the karyotypes of D, M and T plants derived from M plants of mutant dms, and their phenotypes were corresponded with their chromosome constitutions. Now it was clear that the mutant dms was resulted from the absence of chromosome 2A. Our data showed that there were 23 2A-specific genes were highly expressed in stamen. However, it can’t exclude other genes on 2A involved in pollen development also. In the case of wheat Ms1 gene on 4BS, it has homoeologs on 4A and 4D. Mutant ms1 is responsible for male sterile phenotype due to that homoeologous Ms1 on 4A and 4D were not expressed (due to methylation) (Wang et al., 2017). Some genes in the list of 23 2A-specific genes could be candidates of key genes responsible for the male sterility, but not exclude other possibilities. The molecular regulatory network can be elucidated well till the key gene/genes have been investigated.

Endo and Gill reported a set of nullisomic, monosomic, trisomic and tetrasomic lines from Chinese Spring (CS) (Endo & Gill, 1996). Among them, the 2A nullisomic line could not be maintained for they were sterile in both sexes. However, a stable self-fertile 2A nullisomic line was obtained from common wheat ‘Abbondanza’. Although the pistils and stamens of the 2A nullisomic line were fertile, its female flower organ developed unwell (Xue et al., 1991). Different genetic backgrounds lead to nullisomic lines of ‘Abbondanza’ are greater vigor and fertility than those of Chinese spring (Ji, Xue & Wang, 1992). Our 2A nullisomic line dms is multi-pistil and male sterile, that is different from the other 2A nullisomic lines from ‘Abbondanza’ and CS (Endo & Gill, 1996). A set of chromosome deletion stocks from CS are reported, five out of the nine chromosome 2AS deletion lines have irregular meioses with many univalents at metaphase I, and they are highly sterile in both sexes, the seed settings of the four chromosome 2AL deletion lines are reduced after selfing. In our study, the meiosis of the pollen development is normal in dms mutant, which is different from the 2A nullisomic lines from CS (Endo & Gill, 1996).

MYB TFs play pivotal roles in plant development and stress response (Verma, 2019; Zheng et al., 2018). Many MYB TFs have been functionally characterized in pollen development of Arabidopsis and rice, such as AtMYB32 (Preston et al., 2004), OsGAMYB (Aya et al., 2009) and OsTDF1 (Cai et al., 2015). In dms, three homologs of OsGAMYB, TraesCS7A01G458700, TraesCS7B01G357900 and TraesCS7D01G446700, expressed less. In rice, GAMYB is essential for pollen development, and it directly binds to promoter of β-KETOACYL REDUCTASE (KAR), a key enzyme essential for fatty acid synthesis. GAMYB activates the expression of KAR and other genes involved in the synthesis of sporopollenin, and are involved in the formation of exine and Ubisch bodies (Aya et al., 2009). Reduced expression of GAMYB might lead to the shriveled and atrophied pollen grains of D plants. Change in the levels of AtMYB32 expression influence pollen development by affecting the composition of the pollen wall in Arabidopsis (Preston et al., 2004). Knocking out OsTDF1 impaired tapetum development, leading to male sterility in rice (Cai et al., 2015). Similarly, some MYB TF genes expressed less in dms, their functions involved in exine formation in wheat needs further research.

Pollen germination is critical for double fertilization in angiosperms (Zhang, He & McCormick, 2009). The polarity of tip-growing pollen tubes is maintained through dynamic association of active Rho GTPases in plants (ROP-GTP) (Li et al., 2018a). Guanine nucleotide exchange factors for ROPs (RopGEFs) catalyze the activation of ROPs and thereby affect spatiotemporal ROP signaling (Gu et al., 2006). Deletion RopGEF mutant has the defects in pollen tube polar growth (Gu et al., 2006). AGC1.5 and AGC1.7 kinases phosphorylate RopGEFs to control pollen tube growth. Loss functions of AGC1.5 and AGC1.7 in pollen tubes results in meandering and depolarized growth morphology (Zhang, He & McCormick, 2009). In summary, the AGC1.5/1.7-RopGEFs-ROPs signaling pathway is involved in pollen germination and tip growth in Arabodopsis (Li et al., 2018a; Huang et al., 2019). In our research, the homologs of AtAGC1.5, AtAGC1.7 and AtRopGEF8 involved in ROP signaling expressed less in dms. Similarly, all the DEGs associated with various aspects of pollen germination related biological processes, such as pollen tube growth, regulation of pollen tube growth, pollen tube development and pollination, expressed less in dms. These data demonstrated that pollen germination and pollen tube growth might be suppressed in dms. Further experiments should be carried out to test the hypothesis.

In our study, all the six chromosome 2As from different cultivars and the genetic backgrounds of the six heterozygotes of the crosses had similar functions. These indicated that the interactions among 2A, 2B and 2D were similar to that in dms. Wheat chromosome 2A has many important agronomy trait genes such as powdery mildew resistance gene PmLK906 (Niu et al., 2008), photoperiod response locus Ppd-A1 (Beales et al., 2007), reduced height (Rht) genes Rht7 (Worland, Law & Shakoor, 1980). Some quantitative trait loci (QTL) for thousand grain weight (Quan, Sean & Sparkes, 2015), floret primordia development (Guo et al., 2017) and grain protein-content (Groos et al., 2003) are also mapped on chromosome 2A. Because the 2A nullisomic line dms derived from Zhoumai 18 is male sterile but female fertile, it can be used as a female parent to cross with other wheat genotypes, so as to construct inter-cultivar chromosome 2A substitution lines. Backcross to dms can construct series lines with highly similar genetic backgrounds but different 2As (Fig. 10), which can be used to evaluate the functions of different 2As and wheat design breeding targeting 2As.

Figure 10 A program for wheat 2A translocation line construction with the D plants of dms.

The modules coloured in red are the genetic backgrounds of chromosome 2A donor, modules coloured in blue are the genetic backgrounds of D plants, and modules coloured in white are the chromosome 2A lost in D and M plants. T0: chromosome 2A donor plants; M1–MN: the plants harbored one donor chromosome 2A; dmsD: the D plants of dms; T1: New normal wheat lines with two donor chromosome 2As and genetic background of dms. The chromosomes in the modules are chromosome 2As. The number above the plants are the percentages of genetic backgrounds from D plants except for chromosome 2As.

Conclusions

We characterized a dwarf, multi-pistil and male sterile mutant dms derived from a widely-grown wheat cultivar Zhoumai 18. Cytological and molecular analyses demonstrated that mutant dms was a novel wheat 2A nullisomic line. Twenty-three stamen and pollen development related genes are identified specifically on chromosome 2A. We put forward a hypothesis on the molecular regulatory network of the sterility trait in dms. dms is a unique germplasm for gene functional study about chromosome 2A and wheat design breeding targeting 2A.

Supplemental Information

Supplemental Information S1 Supplementary Figures 1-9

Click here for additional data file.

Table S1 The Sequence Read Archive (SRA) data of different wheat tissue type

Click here for additional data file.

Table S2 The primer sequence, the expected products size, and the chromosome location of SSR markers used in this study

Click here for additional data file.

Table S3 Oligonucleotide probes used for FISH

Click here for additional data file.

Table S4 The DNA sequences of the primers used in qRT-PCR for mRNAs

Click here for additional data file.

Table S5 Distribution of the polymorphic SNPs between stamens of T and D plants on 21 pairs of wheat chromosomes

Click here for additional data file.

Table S6 The reads distribution among young spikes of T, M and D plants on 21 wheat chromosomes

Click here for additional data file.

Table S7 The homeologous gene pair between genome 2A and the other genomes

Click here for additional data file.

Table S8 The list of chromosome 2A-specific genes

Click here for additional data file.

Table S9 The chromosome 2A and stamen specifically expressed genes

Click here for additional data file.

Table S10 The DEGs (FC ≥ 4) between stamens of T and D plants

Click here for additional data file.

Table S11 The putative DEG TFs between T-ST and D-ST

Click here for additional data file.

Supplemental Information S2 The full-length uncropped blots (Figure 4)

Click here for additional data file.

Supplemental Information S3 The raw data for qRT-PCR

Click here for additional data file.

We thank Mr. Feiyang Zhao, Northwest Agriculture & Forestry University, for instruction in drawing Fig. 3.

Additional Information and Declarations

Competing Interests

Author Contributions

Field Study Permissions

Data Availability

The authors declare there are no competing interests.

Zhixin Jiao conceived and designed the experiments, performed the experiments, analyzed the data, authored or reviewed drafts of the paper, and approved the final draft.

Xinxin Zhu and Huijuan Li performed the experiments, analyzed the data, prepared figures and/or tables, and approved the final draft.

Zhitao Liu analyzed the data, prepared figures and/or tables, and approved the final draft.

Xinyi Huang, Nan Wu and Junhang An analyzed the data, authored or reviewed drafts of the paper, and approved the final draft.

Junchang Li performed the experiments, analyzed the data, authored or reviewed drafts of the paper, and approved the final draft.

Jing Zhang, Yumei Jiang and Qiaoyun Li performed the experiments, prepared figures and/or tables, and approved the final draft.

Zengjun Qi and Jishan Niu conceived and designed the experiments, authored or reviewed drafts of the paper, and approved the final draft.

The following information was supplied relating to field study approvals (i.e., approving body and any reference numbers):

Permission prior to accessing the field site was provided by Haolong Yang, administrator at the Henan Seed Company.

The following information was supplied regarding data availability:

The sequence data is available at NCBI BioProject: PRJNA612424.

Raw data are available in the Supplemental Files.

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
