# Peer review of "Cytological and molecular characterizations of a novel 2A nullisomic line derived from a widely-grown wheat cultivar Zhoumai 18 conferring male sterility"

_PeerJ, doi:10.7717/peerj.10275_

## Round 0.1 · original submission · Minor Revisions

We have received the comments from three reviewers, all of them recommended that revision is needed.

Reviewer 1 ·

Basic reporting

The article is well written in unambiguous professional English.
Sufficient background along with references are provided.
The article is well presented in terms of article structure etc. With relevant results and hypothesis.

Experimental design

The research in the manuscript is original and within the aims and scope of this journal. The research question is well defined with relevance to wheat genetic research. And although similar in aims to other studies, it adds to those studies and shows different unique outcomes.
The investigation is thorough and technically sound and methods are described in sufficient detail.

Validity of the findings

All the relevant data has been provided and are robust and sound. Conclusions are well stated and linked to the research question. Some modifications to the conclusions are suggested below.

Additional comments

The manuscript titled 'Cytological and molecular characterizations of a novel 2A nullisomic line derived from a widely-grown wheat cultivar Zhoumai 18 conferring male sterility' details the characterization of a novel wheat line deficient in chromosome 2A. Although, 2A nullisomics have been previously reported for wheat (Xue et al., 1991 and 1992), the current manuscripts divulges some previously unknown functions of chromosome 2A.

The line characterized in this manuscript was previously identified by the same authors as a mutant that displayed three pleiotropic phenotypes - male sterility, multiple pistils and reduced plant height. In this manuscript the authors present further data that shows the absence of chromosome 2A (nullisomic 2A) as the main cause for the phenotypes observed. The authors have generated molecular and cytological data including, SNP association, SSR markers and chromosome painting towards the characterization of this mutant. This suggests that original mutant isolated was a spontaneous monosomic line for chromosome 2A.

Further, the authors have made an effort to dissect the molecular basis of male sterility through identification of anther-specific genes on chromosome 2A. Twenty-three genes were identified as candidates controlling anther and pollen development.

The manuscript advances the knowledge of wheat genomics and genetics and shows the various developmental aspects of wheat that chromosome 2A is involved in. Identification of 2A located anther-specific genes furthers the understanding of 2A genes involved in male fertility in wheat. therefore this manuscript is suitable for publication in PeerJ.

However, several revisions are required before the manuscript can be accepted.

1) As the authors mention in the discussion, several chromosome 2AS internal deletions in CS genotype show male sterility including due to meiosis defects, more information is needed on the developmental defects in the 2A nullisomic. Fig S2 should be expanded to include earlier stages of meiosis.

2) In the proposed model, two points should be clarified
i) Is GAMYB homolog (TraesCS7B01G357900) anther-specific in wheat? If it is anther-specific a QPCR panel in figure 8 should be included. If it is not anther-specific then the emphasis in model (fig 9) should be de-emphasized.
ii) Figure 8 includes expression of pollen germination-related genes, however, since pollen germination stages were not analyzed and since pollen are shown to be aborted at microspore stage in D mutants (Fig S2), pollen germination genes should not be expressed at the stages investigated particularly in D mutants. Therefore, the DFGs observed for pollen germination could be homologous genes but not involved in pollen germination. This should be further looked into and clarified in the revised manuscript. The hypothetical model should be accordingly adjusted.

3) In line 289-291, the statement 'Tissue specific expression analysis showed that 188 genes on chromosome 2A were stamen specifically expressed genes (Fig. 6A)' should be rechecked in the text and figure.

Other minor comments:

1) It appears that the term 'proprietary' in line 50 is not appropriate and should be changed to 'novel'. if any intellectual property has been issued or applied for these genes then it should be mentioned.
2) Also consider changing the term 'wizened' to other suitable words such as 'shriveled'
3) Abbreviation 'TF' has been used at several places but it not described in the manuscript
4) In figure S5 all the figures should be arranged in a developmental order youngest to oldest (for anthers as well).
5) In table 2 check the description under F1 (Phenotype and observed lines at F2 (number))

Reviewer 2 ·

Basic reporting

In this study, authors described characterization of a 2A nullisomic line derived from Zhoumai 18 which is male sterile. The language is clear and professional; the methods used to address the hypothesis are appropriate, although some experiments were not necessary, for example, the SSR marker analysis. It seems authors did the cytology analysis last; other analysis was done before the cytology analysis. It is reasonable to include all the experimental data to support the hypothesis. The figures were relevant and raw data were supplied. However, two conclusions drawn from the study were a bit arbitrary; there is not enough evidence to support them. The first, the less expression of MYB transcription factor in D plant might lead to the wizened pollen grains (male sterility) of D plants. This hypothesis was drawn in Figure 9. It seems they excluded other TEs involvement. The results didn’t show how much less expression and how significant it was for MYB. Table_S11 showed differentially expressed TFs. Could authors explain why MYB but not others involved in male sterility/fertility? A few literatures demonstrated that some MYB involved in pollen development. Some other TEs listed in Table_S11 could also be involved in pollen development or plant cuticle biosynthesis, such as C2H2, bHLH, and AP2/ERF (indeed, in Table 3, C2H2 and bHLH were listed). Second, “there were 23 chromosome 2A proprietary genes were involved in stamen and pollen developments” (copied from the abstract). This sentence was not quite appropriate, although data showed that there were 23 2A-specific genes were highly expressed in stamen. It can’t exclude other genes on 2A might be also involved in pollen development. In the case of wheat Ms1 gene on 4BS, it has homoeologs on 4A and 4D. Mutant ms1 is responsible for male sterile phenotype due to that homoeologous Ms1 on 4A and 4D were not expressed (due to methylation). Overall, the discussion part could be improved. Some genes in the list of 23 2A-specific genes could be candidates of key genes responsible for the male sterility, but not exclude other possibilities. Until the key gene/genes are identified, the molecular regulatory network will be elucidated.

Below is the list for minor revisions.

Line 137, “the fifteen bulks” should be “the fifteen samples”.

Line 161, “Clarke, 2013” should be Clarke, 2009”.

Line 227-228, “between stamens of T and D plants” should be “between T and D plants”.

Line 248, “In a word” should be “In another word”.

Line 299, if you use “TF” first time, it should be “transcription factor (TF)”.

Line 335, “gravely” should be “greatly” or “seriously”?

Line 348, “they all were male fertile but female sterile”. It is not correct. In Endo and Gill (1996)’s paper, 2A nullisomic and some deletion lines were described as they were sterile in both sexes.

Line 361, “Verma, 1996” should be “Verma, 2019”.

Line 391, “both the six chromosome 2As….” Should be “all the six ….”.

Line 396, “Same” should be “Some”.

Table 3, reference “Fu et al. 2018” was not in the reference list.

Table 4, two references “Li et al. 2018”, they should be 2018a, 2018b?

Please check the format of references cited in the text; they were not constant, sometime as “Li et al. 2018”, sometime “Li et al., 2018”. Please check the Journal’s guidance.

Fig. S5, samples should be described more clearly.

Experimental design

Experimental design to address the hypothesis is appropriate.

Validity of the findings

Some conclusions need to be revised. See basic reporting.

Additional comments

no comment.

·

Basic reporting

This paper is fine in these aspects, except for English expression.

I have attached a version of the Abstract and Introduction of the paper where I have corrected/made suggestion in word with track-changes and saved as a pdf. This will give the authors the ideas of what should be done in the rest of the paper.

Experimental design

Once again, this papers is fine in these technical aspects.

Validity of the findings

Also fine.

Additional comments

This paper is fine in all technical aspects, and very interesting, the researchers have done a nice job. English expression needs some work.

I have attached a version of the Abstract and Introduction of the paper where I have corrected/made suggestion in word with track-changes and saved as a pdf. This will give the authors the ideas of what should be done in the rest of the paper.

---

## Round 0.2 · accepted · Accept

The authors have made changes according to comments from reviewers. The manuscript can be accepted for publication.